# Lab-Scale Study of Temperature and Duration Effects on Carbonized Solid Fuels Properties Produced from Municipal Solid Waste Components

**DOI:** 10.3390/ma14051191

**Published:** 2021-03-03

**Authors:** Kacper Świechowski, Paweł Stępień, Ewa Syguła, Jacek A. Koziel, Andrzej Białowiec

**Affiliations:** 1Department of Applied Bioeconomy, Wrocław University of Environmental and Life Sciences, 37/41 Chełmońskiego Str., 51-630 Wrocław, Poland; kacper.swiechowski@upwr.edu.pl; 2Institute of Agricultural Engineering, Wrocław University of Environmental and Life Sciences, 37/41 Chełmońskiego Str., 51-630 Wrocław, Poland; pawel.stepien@upwr.edu.pl; 3Department of Agricultural and Biosystems Engineering, Iowa State University, Ames, IA 50011, USA; koziel@iastate.edu

**Keywords:** CO_2_-assisted pyrolysis, organic waste, waste conversion, thermal treatment, waste to energy, waste to carbon, regression models, waste recycling, municipal waste, circular economy

## Abstract

In work, data from carbonization of the eight main municipal solid waste components (carton, fabric, kitchen waste, paper, plastic, rubber, paper/aluminum/polyethylene (PAP/AL/PE) composite packaging pack, wood) carbonized at 300–500 °C for 20–60 min were used to build regression models to predict the biochar properties (proximate and ultimate analysis) for particular components. These models were then combined in general models that predict the properties of char made from mixed waste components depending on pyrolysis temperature, residence time, and share of municipal solid waste components. Next, the general models were compared with experimental data (two mixtures made from the above-mentioned components carbonized at the same conditions). The comparison showed that most of the proposed general models had a determination coefficient (R^2^) over 0.6, and the best prediction was found for the prediction of biochar mass yield (R^2^ = 0.9). All models were implemented into a spreadsheet to provide a simple tool to determine the potential of carbonization of municipal solid waste/refuse solid fuel based on a local mix of major components.

## 1. Introduction

Waste generation is an inherent element of human activity and economic development. The Organization for Economic Co-Operation and Development (OECD) estimates increases in waste production by 0.69% for each 1% of the gross domestic product (GDP) [1]. Proper municipal solid waste (MSW) management becomes more difficult every year as new materials and products are introduced. For example, new packing materials are increasingly composite materials (the mixture of several materials designed for desired properties). Composite waste materials are challenging to effectively recycle when the mechanical separation is applied [2]. Thus, new technologies are needed to sustainably treat MSW in the context of zero landfilling and circular economy goals.

The MSW is a heterogeneous mixture of various materials generated in households, public, commercial and industrial sectors [3]. Many factors affect the MSW composition (e.g., economic development, the level and type of urbanization, culture, law, climate, season, environmental awareness of citizens). The MSW composition is highly variable with time, but in general, organic waste constitutes up to 83% [4]. The principal organic components of MSW are food waste, green waste (e.g., grass clippings, leaves, branches), wood, paper, carton, rubber, and the most abundant plastics [5]. In the last 20 years, plastics production doubled to ~335,000,000 Mg globally and is estimated to exceed >600,000,000 Mg in the next ten years [6].

The EU Directive 2008/98/EC proposed the waste management hierarchy, promoting waste prevention, followed by reuse, and then recycling [7]. If recycling is not possible, the recovery should be used (e.g., incineration with energy recovery), and finally, residual waste disposal [7]. The Circular Economic Action Plan has recently been adopted by the European Commission, aiming at climate neutrality by 2050. One of the targets is to use waste instead of raw materials [8].

To date, a common practice in the EU is the initial separation of MSW fractions directly at the source (e.g., separation of paper, plastic, biodegradable waste, glass, and residual waste at households). After collection, these pre-separated fractions are processed in a waste mechanical sorting plant. The residual wastes usually are treated in so-called mechanical biological treatment plants (MBT) [9,10]. The main goals are to (i) recycle high-value materials, (ii) to produce refuse-derived fuel (RDF) from waste that cannot be recycled, (iii), and to stabilize biodegradable waste before its disposal in landfills [11].

RDF is a fuel made from various types of materials by its homogenization (shredding and mixing), removal of not flammable waste (e.g., glasses, metals, stones), removal of chlorinated waste, and drying if needed (e.g., by bio-drying). The RDFs are used for heat and electricity production in specially designed incineration plants or in cement kilns to substitute fossil fuels. RDF has to meet standards that depend on user requirements. More valuable RFD has a higher calorific value, lower ash content, and moisture. RDF fuel quality can be improved by mechanical treatment. However, the thermal conversion of RDFs is growing [12,13,14].

The thermal conversion of waste in the absence or limited amount of oxygen includes torrefaction, hydrothermal carbonization, low-temperature pyrolysis, pyrolysis (slow, intermediate, fast), and gasification. Torrefaction and low-temperature pyrolysis are favorable means to upgrade the RDF quality, as these processes lead to a significant increase of carbonized solid fuel (CSF) calorific values [15,16]. The higher temperatures (e.g., pyrolysis at ~600 °C) leads to the increased production of gases and liquid fractions at the expense of solid fraction and the ultimate loss of the CSF’s calorific value [14,17,18].

The preferred low-temperature pyrolysis occurs at 300–550 °C and results in biochars (e.g., CSF when RDF is a feedstock) and pyrolysis gases, from which the liquid fraction can be separated [19]. The CSF mainly consists of carbon. The pyrolytic gas consists of H_2_, CO, CO_2_, CH_4_, and other low-molecular-weight hydrocarbon gases, whereas the liquid fractions are a mixture of various oils that can be processed into useful chemicals. There is also possible to use pyrolysis gases (with or without liquid separation) to provide energy to the pyrolysis process by its incineration. The optimal temperature, heating rate, process duration, and size of particles depend on the processed material and its processing goals. For CSF production, a relatively slow heating rate is recommended [20,21].

Mass yield (MY), energy densification ratio (EDr), and energy yield (EY) are the key parameters characterizing the production of CSF for energy purposes. The mass yield shows how much CSF will be obtained after pyrolysis in relation to the initial mass. The energy densification ratio describes how much energy was densified in CSF in relation to the calorific value of the feedstock. The energy yield is the amount of energy remaining in the material after processing and is given in %.

Proximate analyses such as moisture content (MC), volatile matter content (VM), fixed carbon content (FC), and ash content (AC) are useful for fuels (MC + VM + FC + AC = 100%) [22,23]. The combustible parts (CP) and volatile solids (VS) are often reported for waste pyrolysis. The CP is determined mainly for MSW (as all organic and inorganic matter that is converted into gas during incineration at 815 °C) [24]. The VS (a.k.a. organic matter content/loss on ignition) consists of organic substances converted into a gas at 550 °C [25]. The calorific value determination is focused on the high heating value (HHV) and low heating value (LHV), wherein typically, the LHV is calculated as a function of HHV, MC, and H content [26].

For the ultimate analysis, the elemental composition is determined (C, H, N, S, and O), where (typically) C + H + N + S + O + AC + MC = 100% [22]. In general, the higher the C and H content, the higher the calorific value. After pyrolysis, most of the carbon stays in CSF, whereas most of the H goes to the gaseous phase. O decreases the calorific value, and N and S are a source of combustion pollutants (NO_x_ and SO_x_) [27].

The inherent variability in MSW composition is the most significant barrier to determining the resulting CSF fuel properties [28]. Each time the raw feedstock material has to be tested, it could be time-consuming and costly. Thus, to minimize resources, methods are used to predict the test material’s properties under different process conditions [29]. Process and biochar properties modeling can be done using data from elemental composition, proximate analysis, or other analyses, which correlate to modeled variables. This modeling can be done using mathematical models (empirical, based on implicit theory, semiempirical, or neural networks). The most commonly used are linear regression models, where empirical data is subjected to regression by the least-squares method. The most well-known regression model is used for HHV determination and is called Dulong’s formula [30]. There are also other formulas for different materials [29] and different input data [31]. The precision of equations based on proximity analysis is lower than that of CHNS or neural network analysis, but the main advantage of these methods is their simple structure and low cost [32].

In this work, we used different approaches to modeling of pyrolysis process and biochar properties. We used experimental data from processed materials and correlated it with process conditions. Conventional modeling uses data from substrate analysis (proximate, ultimate, other) and then correlates it with modeled variable. As a result, our models do not require time-consuming and costly analyses of substrates as input data.

The objective of this work was to develop empirical models for the CSF quality that incorporate the effects of low-temperature pyrolysis conditions (duration (t), temperature (T) for the mixture of the main constituents of MSW. The work’s main result was a calculation tool (Excel-based file for modeling) that predicts CSF properties from waste pyrolyzed at 300–500 °C at durations up to 60 min without the need to analyze input material characteristics.

## 2. Materials and Methods

The quantified parameters included MY, EDr, EY, MC, OM, AC, CP, and C, H, N, and S content. The main MSW components included: carton, fabric (cotton t-shirt), kitchen waste, paper, plastic, rubber, paper/aluminum/polyethylene (PAP/AL/PE) composite packaging packs, and wood. A detailed description of methods is provided elsewhere [14]. ‘CSF’ replaces the name ‘biochar’ used for the thermally processed material in similar conditions for the rest of the manuscript.

First, measured properties of particular waste components were subjected to four regression models (models I–IV). The best-fitted model was chosen for each parameter and each waste component among these four models, based on the determination coefficients (R^2^) and Akaike information criterion (AIC). Then, the best-fitted models for particular waste and parameters were combined in one general model. Overall, 11 general models were developed, one for each studied parameter. Next, the general models were compared with experimental data. The experimental data included two (example) RDF mixtures (with a known share of particular components) carbonized at the same condition conditions as the individual components.

### 2.1. Materials

The main MSW components were: carton (grey carton), fabric (cotton t-shirt), kitchen waste (vegetables, 41.6% (carrot 13.86%, potato 13.86%, salad 13.86%); banana peel, 29.7%; basic food (pasta 7.43%, rice 7.43%, bread 7.43%); chicken, 0.2%; eggshells, 4%; and walnut shells, 2.2% by weight), paper (office paper), plastic (polyethylene foil), rubber (car inner tube), PAP/AL/PE composite packaging pack (Tetra Pak, Lund, Sweden, Lausanne, Switzerland), and wood (pruning tree branches).

The RDF mixtures were used for validation of the general models. The mixture consisted of the following fractions: RDF—carton, 9.64%; fabric, 6.20%; kitchen waste, 4.02%; paper, 9.64%; plastic, 34.23%; rubber, 9.6%; PAP/AL/PE composite packaging pack, 12.22%; wood, 14.45%; RDF 2—carton, 8.57%; fabric, 9.54%; kitchen waste, 7.10%; paper, 8.57%; plastic, 45.24%; rubber, 7.71%; PAP/AL/PE composite packaging pack, 5.81%; wood, 7.46%. More detailed information is presented elsewhere [14].

### 2.2. Methods

#### 2.2.1. Low-Temperature Pyrolysis

Each main component and RDF mixtures were carbonized at 300–500 °C (interval 20 °C, and residence time 20–60 min (interval 20 min) using a muffle furnace (Snol 8.1/1100, Utena, Lithuania). CO_2_ was supplied (2.5 dm^3^∙min^−1^) to provide inert conditions. Samples were heated at 50 °C∙min^−1^ to the setpoint temperature. After the end of setpoint residence time (20–60 min), the furnace was cooled down by itself while the CO_2_ was still provided to prevent the self-ignition of carbonized solid fuels. The CSF samples were then removed from the furnace when the temperature was <200 °C [14].

#### 2.2.2. Regression Modeling

Raw data used for the regression model come from previous work [14]. In short, for each produced CSF analyzed for MY, EDr, EY, C, H, N, and S content, 1 repetition was done; whereas, for MC, OM, AC, and CP, three repetitions were completed. Therefore, models of MY, EDr, EY, C, H, N, and S for each material was made using 33 measurement points, whereas MC, OM, AC, and CP were made using 99 measurement points.

Properties of each waste component were subjected to four regression models using the method of least squares to estimate intercept (a_1_) and regression coefficients (a_2_–a_6_):Model I—linear equation y(T,t) = a_1_ + a_2_T + a_3_ × t;Model II—second-order polynomial equation y(T,t) = a_1_ + a_2_ × T + a_3_ × T^2^ + a_4_ × t + a_5_ × t^2^;Model III—factorial regression equation y(T,t) = a_1_ + a_2_ × T + a_3_ × t + a_4_ × T × t;Model IV—response surface regression equation y(T,t) = a_1_ + a_2_ × T + a_3_ × t + a_4_ × T^2^ + a_5_ × t^2^ + a_6_ × T × t.
where:y(T,t)—the variable that depends on process temperature and time.T—low-temperature pyrolysis process temperature (°C), (300–500 °C).t—low-temperature pyrolysis process residence time (min) (20–60 min).a_1_—intercepta_2_–a_6_—regression coefficients

The best-fitted model was chosen for each parameter and each waste component among models I–IV, using R^2^ and AIC, respectively [33].
(1)$$R^{2}=\frac{\sum_{i=1}^{n}{\left(\hat{y_{i}}-\bar{y}\right)}^{2}}{\sum_{i=1}^{n}{\left(y_{i}-\bar{y}\right)}^{2}}$$
where:
$R^{2}—$determination coefficient;i—repeated observations;$\hat{y_{i}}$—value of the dependent variable predicted by the regression model;$\bar{y}$—mean value of the dependent variable (measured);$y_{i}$—value of the dependent variable (measured).


(2)$$\mathrm{AIC}=n\cdot \ln(\overset{n}{\underset{i=1}{\sum }}e_{i}^{2})\;+\;2\cdot K$$
where:AIC—a value of Akaike analysis;n—the number of measurements;e—the value of the residuals of the model (defined as the difference between predicted and experimental value);K—number of regressions coefficients (including the intercept).

The a_1_, a_2_–a_6_, and R^2^ were calculated using StatSoft software Statistica 13.3 (TIBCO Software Inc., Palo Alto, CA. USA). The best-fitting model for each variable was determined as follows. First, the model (I–IV) with the highest R^2^ was chosen. If several models had similar R^2^, the model with the lowest AIC was chosen as the best-fitted. The R^2^ quantifies how well models match to data, whereas the AIC points to the simpler model with similar matching.

#### 2.2.3. General Model

The best-fitted models for each main waste and parameters were combined into one general model:(3)$$y_{\mathrm{CSF}\;}\left(T,t\right)=\frac{\sum_{i}^{n}\left(y_{i}\left(T,t\right)\cdot {\%\mathrm{share}}_{i}\right)}{\sum_{i}^{n}\left({\%\mathrm{share}}_{i}\right)}$$
where:

$y_{\mathrm{CSF}}\left(T,t\right)$—the estimated value of the studied parameter of CSF from a mixture of MSW/RDF at T & t conditions, MJ∙kg^−1^;$y_{i}\left(T,t\right)$—the estimated value of the studied parameter of i-CSF from individual MSW/RDF component under T & t conditions, MJ∙kg^−1^;${\%\mathrm{share}}_{i}$—percentage mass share of i-CSF from individual MSW/RDF component in the total mass of CSF from MSW/RDF mixture, %.

Overall, 11 general models were developed, one for each studied parameter. Next, general models were compared with experimental data of two RDF mixtures (with a known share of particular components) carbonized at the same condition conditions as the individual components, RDF 1 and RDF 2. A linear correlation (R) and R^2^ were used to compare general models.

## 3. Results and Discussion

### 3.1. Regression Models

#### 3.1.1. Mass Yield of CSF

For all MY models (except rubber), model IV had the highest R^2^ and the lowest AIC; therefore, model IV was assumed as the best fit. The best model for rubber was model II (Table A1 in Appendix B). The influence of low-temperature pyrolysis temperature and residence time on CSF’s mass yield of best-fit equations are shown in Table 1.

The MY of the CSF from the carton decreased with increasing T and t from 92% to 32%, model R^2^ = 0.77. The CSF results from the fabric showed that as the T increased, the MY decreased from 98% to 18%, model R^2^ = 0.78. The MY of the CSF from kitchen waste decreased with increasing T and t from 85% to 37%, the model R^2^ = 0.85. The paper pyrolysis results showed that as the T and t increase, the MY decreased from 91% to 37%, model R^2^ = 0.79. The plastic subjected to the pyrolysis was characterized by a high drop in MY with an increase in T (up to 500 °C) and t (up to 60 min) from 99.9% to 8%, model R^2^ = 0.72. Rubber was characterized by a linear decrease in MY with an increase in T. The increase in t did not show a significant influence on MY. The MY of CSF from rubber decreased from 99% to 40%, model R^2^ = 0.85. The MY of the pyrolyzed PAP/AL/PE was characterized by a linear decrease with increasing T from 96% to 25%, model R^2^ = 0.85. Wood waste was characterized by decreased MY with increasing T from 92% to 32%, model R^2^ = 0.81.

Each material was characterized by a decreasing MY trend caused by increasing T and t. In general, the proposed models had a high R of ~0.80. The lowest R^2^ was found for plastic due to outliers at 500 °C at 60 min. These outliers are probably a result of the lack of secondary reactions. The plastic (polyethylene) was pyrolyzed in the muffle furnace with the constant inert gas flow (CO_2_). Therefore, the vapored liquids were removed from the reactor. As a result, these liquids could not generate biochar in the secondary reactions that occur at >440 °C and >90 min [34].

The correlation between the experiment results and the model data for MY is shown in Figure 1. The confidence interval for correlation was 95% (dotted lines). The results showed that the proposed general model had correlations of 95% with the experimental data for RDF 1 and RDF 2, respectively. Obtained results indicate that the proposed model concerning pyrolysis’ T, t, and knowledge of RDF composition could be useful to predict the MY produced from pyrolyzed RDF.

#### 3.1.2. Energy Densification Ratio of CSF

For all materials, the influence of low-temperature pyrolysis T and t on EDr of CSF was best described by model IV, which had the highest R^2^ and the lowest AIC (Table A1). These models’ equations were summarized in Table 2.

The EDr of CSF produced from the carton did not increase significantly with the T and t and the best-fitted model had R^2^ = 0.16 (Table 2). The CSF produced from cotton was characterized by increasing EDr as the T increased, and the best-fitted model had R^2^ = 0.55 (Table 2). The EDr of CSF produced from kitchen waste did not increase significantly when the process T and t increased, and the determination coefficient for the best-fitted model was R^2^ = 0.55. The CSF produced in the paper was characterized by a decrease in the EDr with the T increase, and the R^2^ for the best-fitted model was R^2^ = 0.41. The EDr of CSF produced from plastic increased as the T increased to 460 °C and then started to decrease. The highest R^2^ for the plastic model was 0.55 (Table 1). The CSF from rubber was characterized by a decrease in the EDr as the T increased, and the R^2^ for the best model was 0.88. The EDr of CSF produced from PAP/AL/PE composite packaging pack increased with T increase up to 460 °C and then started decrease. The R^2^ for the best model was 0.73. The CSF produced from wood was characterized by an increase in the EDr as the T increased, and the R^2^ for the best-fitted model was 0.82 (Table 2).

The EDr results showed that for fabric, kitchen waste, PAP/AL/PE composite packaging pack, and wood, the higher T, the higher energy becomes concentrated in CSF even up to EDr = 1.4. The opposite tendency was for plastic and rubber, for which pyrolysis led to a decrease in EDr < 1. This means that the CSF had less energy than the substrate used for its production. This phenomenon might be due to the higher rate of AC increase than FC rate, thus, lower EDr in these CSFs.

The correlation between the experiment and the model data for EDr is shown in Figure 2. The correlation between the proposed general model and experimental data was 85% for RDF blend 1 and 87% for RDF blend 2, respectively. It means that based on pyrolysis’ T, t, and knowledge of RDF composition share, it is possible to predict the energy densification ratio of RDF after its carbonization.

#### 3.1.3. Energy Yield of CSF

For all EY, model IV turned out to be the best-fitted model, except for plastic and rubber, for which model II was better (Table A1). These models’ equations presenting the influence of low-temperature pyrolysis T and t on the EY were shown in Table 3.

The EY of CSF from carton decreased with increasing T and t from 97% to 35% model R^2^ = 0.76. Pyrolysis of the fabric was characterized by a decrease in EY along with an increase in T and t from 96% to 26%, model R^2^ = 0.81. The CSF from kitchen waste was characterized by decreased EY with an increase in T and t from 93% to 47%, model R^2^ = 0.81. The EY of CSF from paper decreased with increasing T and t from 99.9% to 28%, model R^2^ = 0.81. The CSF from plastic was characterized by a decrease in EY along with an increase in T and t from 98% to 8%, model R^2^ = 0.24. The EY of CSF from rubber decreased with increasing T and t from 99% to 20%, model R^2^ = 0.85. The EY of the CSF made from the PAP/AL/PE composite packaging pack was characterized by a decrease with increasing T from 99% to 25%, model R^2^ = 0.78. Wood pyrolysis was characterized by a decrease in the EY along with an increase in T and t from 98% to 41%, model R^2^ = 0.79.

The results showed that for all MSW/RDF components, the EY decreases with the increase of T and t. The decrease in EY of solid fractions with increasing T is typical for most materials converted thermally [35,36]. Despite increased HHVs, the EY decreased due to the significant decrease in MY of the CSFs [37].

The correlation between the experiment and the model for EY is shown in Figure 3. The confidence interval was 95%. The results showed that the proposed model explains 74% and 61% of data variability for RDF blend 1 and RDF blend 2, respectively. The obtained results indicate that it is possible to predict EY with accuracy over >60% based on pyrolysis T, t, and RDF composition share.

#### 3.1.4. Moisture Content of CSF

For most MC models, model IV was the best fitted, but for fabric and PAP/AL/PE composite packaging pack, model II was better, and for the wood model I (Table A2). The best-fitted equations of influence of T and t on the MC of CSF are presented in Table 4.

The MC of CSF from carton ranged from 0.51% to 4.19%, with no linear relationship. The highest MC was observed for CSF produced at 300 °C and 60 min, while the lowest for CSF produced at 20 min and 300–440 °C, model R^2^ = 0.11. The CSF from cotton had an MC of 0.16% to 7.86%. The highest MC was found in CSF produced at 60 min & 440–500 °C, while the lowest at 20 min at 300–320 °C, model R^2^ = 0.13. The MC of CSF from kitchen waste ranged from 0.36% to 7.22%. The highest MC was found in CSFs produced at 20–40 min and 440–500 °C, model R^2^ = 0.10. CSF made from the paper had an MC of 0.34% to 6.23%. The highest MC was observed in CSF produced at 300 °C and the lowest at 360–420 °C, model R^2^ = 0.21. The results of the MC of CSF made from plastic ranged from 0.01% to 1.36%. The highest MC was in CSF produced at 300 °C, whereas the lowest in CSF produced at 400–460 °C, model R^2^ = 0.35. CSF made of rubber had an MC of 0.1% to 2.37%. The highest MC was found in CSF produced at 440–500 °C in 60 min and the lowest at 480°C in 40 min, model R^2^ = 0.05. The MC of CSF from PAP/AL/PE composite packaging pack waste was between 0.18% and 5.21%. The highest MC was observed for CSF produced at the lowest and highest T (at 300 °C and 500 °C), while the lowest MC at 380–420 °C, model R^2^ = 0.49. The results of MC of CSF from wood ranged from 0.08% to 6.25%. The highest MC was found in CSF produced at 480–500 °C, while the lowest MC was found in CSF produced at 300 °C, model R^2^ = 0.05.

The proposed model of MC for CSF had R^2^ ranging from 0.05–0.49. The reason for that could be that CSF samples were generated and stored for a long time before MC determination. Similar results (no tendency) were found in our previous work about torrefaction [16]. For that reason, additional tests should be done to reveal the cause of that phenomenon.

The correlation between the experiment and the model data for MC is shown in Figure 4. The results showed that the proposed model could not predict the MC of CSF, and it explains only 2% and 10% of data variability for RDF blend 1 and RDF blend 2, respectively. The MC of CSF was characterized by high scatter. Caution should be exercised with the proposed model for MC.

#### 3.1.5. Organic Matter Content of CSF

For most OM models, model IV was the best fitted, except fabric and wood, for which the best model was model II (Table A2). The equations of influence of low-temperature pyrolysis T and t on the OM of CSF are shown in Table 5.

The OM in CSF from carton ranged from 44% to 83%. As the T and t increased, OM’s content in CSF decreased, model R^2^ = 0.78. CSF produced from the fabric contained ~99.5% of OM. The increase of T and t slightly led to a decrease in OM, model R^2^ = 0.28–0.31. The OM in CSF produced from kitchen waste ranged from 50% to 83%, model R^2^ = 0.49. The results of the OM of CSF produced from paper ranged from 29% to 78%. As the T and t increase, the OM in CSF decreased, model R^2^ = 0.72. CSF made from plastic contained OM in the range from 4% to 87%, model R^2^ = 0.69. The content of OM in CSF made of rubber ranged from 47% to 86%. As the T and t increase, the OM in CSF decreased, model R^2^ = 0.84. CSF produced from PAP/AL/PE composite packaging pack waste had OM between 49% and 87%. As the T and t increase, the OM in CSF decreased, model R^2^ = 0.83. The OM in CSF made from wood ranged from 83% to 98%. In general, the increase in T led to a decrease in OM, model R^2^ = 0.49.

The proposed models of OM in CSF made from MSW components had low (e.g., fabric, wood) and high R^2^ (eq. carton, rubber), respectively. Despite the low R^2^ for fabric and wood, the models should reflect the general trend in the experimental data because the OM showed a decreasing linear relationship with an increase in T, and the differences in OM for CSF produced at 300 and 500 °C were low. The general trend in the decrease of OM in CSF with an increase of T and t was observed. This finding is in agreement with previous work [12]. The OM decrease is a result of OM volatilization. During pyrolysis, the processed material molecules are thermally decomposed (breaking into smaller ones), vaporized, and off-gased. The volatilization is also dependent on the pyrolysis T, and it increases when T increase.

The correlation between the experiment and the model data for OM is shown in Figure 5. The results showed that the proposed model explains 77% of the data variability for RDF 1 and RDF 2, respectively. The confidence interval was 95%. The largest concentration of results is in the range 80–85% for experimental data and 75–85% for models, which means the model lowers the value slightly. The results indicate that based on the assumptions of pyrolysis T, t, RDF components share, it is possible to predict the OM in carbonized RDF.

#### 3.1.6. Ash Content of CSF

For most AC models, model IV turned the best fitted. The best-fitted models for fabric and wood were I and II, respectively (Table A2). These equations are shown in Table 6.

The AC of CSF made from cartons ranged from 10% to 40%. As the T and t increased, the AC increased, model R^2^ = 0.84. The CSF produced from the fabric had an AC of ~1%, model R^2^ = 0.35. The AC of CSF produced from kitchen waste ranged from 8% to 29%, model R^2^ = 0.40. The CSF made from paper contained AC between 3% and 41%, model R^2^ = 0.63. AC in CSF made of plastic was between 8% and 58%, model R^2^ = 0.77. The CSF made of rubber contained AC between 11% and 47%, model R^2^ = 0.78. The AC of CSF made from PAP/AL/PE composite packaging pack ranged from 7% to 45%, model R^2^ = 0.86. The CSF made from wood had an AC ranging from 0.7% to 12%, model R^2^ = 0.32.

For each material, as the T and t increased, the AC increased. The observed trend is typical for CSF, the OM is removed from the processed material in gases and tar while the inorganic fraction remains [38,39]. The biggest surprise was the increase of AC for CSF made from plastic at 500 °C at 60 min, where its content increased almost to 60%. The reason for that could be that the reactor was flushed with CO_2_ gas, and as a result, no secondary reaction could form a char from liquids [34]. The other explanation for that phenomenon may be the Boudouard reaction (CO_2_ + C = 2CO), where char could be transformed into CO. Nevertheless, the Boudouard reaction can occur at higher temperatures >700 °C [40]. The partial oxidation (incineration) of the plastic is rather unlike because all RDF components were pyrolyzed at the same time, and the other types of samples did not show the same tendency.

The correlation between the experiment and the model data for AC is shown in Figure 6. The results showed that the proposed model explains 66% and 69% of data variability for RDF blend 1 and RDF blend 2, respectively. The confidence interval was 95%. The obtained results indicate that based on the assumptions of pyrolysis T, t, and knowledge of RDF composition, it is possible to predict the AC of carbonized RDF.

#### 3.1.7. Combustible Parts of CSF

For most CP models, model IV was the best fitted, but for fabric and wood better were models I and II, respectively (Table A2). The equations describing the influence of low-temperature pyrolysis T and t on the CP of CSF are summarized in Table 7.

The CP in the CSF formed from the carton ranged from 59% to 90%, model R^2^ = 0.84. The CSF from the fabric contained 97% to 99% of the CP, model R^2^ = 0.35. The content of CP in CSF produced from kitchen waste ranged from 71% to 89%, model R^2^ = 0.40. The CSF made from paper contained 58% to 96% of CP, model R^2^ = 0.63. The CP in CSF made from plastic ranged from 42% to 91%, model R^2^ = 0.77. The CSF made of rubber contained 52% to 89% of CP, R^2^ = 0.78. The content of CP in the CSF made from PAP/AL/PE composite packaging pack ranged from 55% to 93%, model R^2^ = 0.86. The CSF from wood contained 89% and 99% of CP, model R^2^ = 0.32.

The decreasing trend of CP with an increase of T and t is typical and very similar to the OM. A decrease of CP in RDF pyrolyzed was previously reported by Stępień et al. [5]. Carbonized RDF resulted in the CP decrease (from 81% to 43%) and AC increase (from 18% to 57%) [5].

The correlation between the experiment results and the model data for CP is shown in Figure 7. The results showed that the proposed model explained 66% and 69% of data variability for RDF blend 1 and RDF blend 2, respectively. The confidence interval was 95%. The obtained results indicate that based on the assumptions of pyrolysis T, t, and RDF composition, it is possible to predict the CP of carbonized RDF.

#### 3.1.8. Carbon Content of CSF

For all tested materials, the C was described best by model IV (Table A3). The equations of influence of low-temperature pyrolysis T and t on the C of CSF are summarized in Table 8.

In general, for CSF from fabric and wood, the increase of T and t increased the C. An opposite trend was observed for plastic and rubber where CSF produced at lower T had higher C. No impact of T and t on the C of CSF made from the carton, paper, and kitchen waste was observed. The C of CSF made from the carton ranged from 39% to 48%, and the highest R^2^ was 0.46. The C of CSF made from fabric ranged from 43% to 73%, and the highest R^2^ was 0.62. The CSF produced from kitchen waste contained 41% to 59% of C, and the highest R^2^ was 0.26. The C of CSF made from paper ranged from 35% to 46%, and the highest R^2^ was 0.24. The CSF made from plastic contained between 13% and 73% of C, and the highest R^2^ was 0.72. The C of CSF made from rubber was in the range of 55% to 85%, and for the best-fitted model, R^2^ = 0.69. The CSF from the PAP/AL/PE composite packaging pack contained between 35% and 62% of C and the best-fitted model had R^2^ = 0.63. The C of CSF made from wood was between 47% and 67%, and the best-fitted model had R^2^ = 0.72.

The trends of C changes os studied materials are in agreement with previous research. In general, decarbonization of the substrate occurs during pyrolysis, and an increase of C content in char compared to the untreated substrate is a result of reduced char mass-the remaining char becomes an increasingly condensed carbon matrix [41]. Muley et al. [41] pyrolyzed cellulose and sawdust at temperatures from 500 °C to 700 °C and found that the C increased from 42% to 79% and 47% to 89%, respectively. In the present study, cellulose materials (carton, fabric, paper) pyrolyzed at 500 °C had C in the range of 44–60% and did not show such a large increase, probably because of too low pyrolysis T. Quite interesting is the observed decrease in C in rubber from 80% to 56%. In the work of Mui et al. [42], waste tire rubber was carbonized at 400–900 °C and showed the opposite trend, where the C increased from 80% to 88% [42]. On the other hand, Acosta et al. [43] showed that scrap tires’ carbonization at 570 °C led to a decrease in C from 86% to 80%. These differences may be a result of the different compositions of carbonized rubber.

The correlation between the experiment and the model data for C is shown in Figure 8. The results showed that the proposed model explained 76% and 74% of data variability for RDF blend 1 and RDF blend 2, respectively. The confidence interval was 95%. The largest concentration of results was 60–65% of C for the experimental and the modeled data. The obtained results indicate that based on pyrolysis T, t, and knowledge of RDF composition, it is possible to predict the C in CSF.

#### 3.1.9. Hydrogen Content of CSF

For H determination in CSF, the best model type was model IV, except for kitchen waste, where model II had the better fit (Table A3). The equations of influence of low-temperature pyrolysis T and t on the H of CSF are summarized in Table 9.

For all pyrolyzed materials, as process T and t increased, the H decrease was observed. The exception was wood, for which the H slightly increased. The H of CSF made of carton ranged from 2.1% to 6.3%, and its content decreased when the process T and t increased. The model’s R^2^ was 0.92. The CSF made from fabric contained from 2.4% to 7.7% of H, and its content decreased when process T and t increased. The model’s R^2^ was 0.79. The CSF produced from kitchen waste contained from 2.1% to 6.2% of H, and its content decreased when process T and t increased. The model’s R^2^ was 0.83. The H content of CSF made from the paper was between 1.6% and 5.2%, and its content decreased when process T and t increased. The model’s R^2^ was 0.88. CSF made from plastic contained from 3% to 12% H. The H content decreased with an increase in process T. The model R^2^ = 0.66. The H content of CSF made from rubber contained 0.05% to 3.45% of H, which decreased with an increase in process T. The model R^2^ was 0.66. The CSF produced from PAP/AL/PE waste composite packaging pack contained 1.5% to 7.9% of H, which decreased with an increase in T. The model R^2^ was 0.72. The H of CSF made of wood ranged from 2.8% to 5.7%, which increased with an increase in process T. The model R^2^ was 0.72.

In general, for all pyrolyzed materials, as the T and t increased, the H decreased. The exception to this observation was wood, for which the H increased, which is opposite to the results of Muley et al. [41], Zeng et al. [44], and Ho Kim et al. [45], where H decreases were observed. Nevertheless, similarly to C, the H also is removed during pyrolysis (dehydrogenation) [46] and its content in char compared to the substrate (increase or decrease) depends on the mass reduction of all components.

The correlation between the experiment and the model data for H content is shown in Figure 9. The results showed that the proposed model explained 69% and 74% of data variability for RDF blend 1 and RDF blend 2, respectively. The confidence interval was 95%. The largest concentration of results is in the range of 8–10% for experimental data and 13–15% for the models, which means that the general model slightly overestimates the H in CSF. The obtained results indicate that based on the assumptions of pyrolysis T, t, and RDF composition, it is possible to predict the H in carbonized RDF.

#### 3.1.10. Nitrogen Content of CSF

For all CSF, the best model for N determination turned out to be model IV. These models IV had the highest R^2^ and the lowest AIC value (Table A3). The equations of influence of low-temperature pyrolysis T and t on the N of CSF are summarized in Table 10.

The N of CSF made of carton ranged from 0.1% to 1.7%, and the model had R^2^ = 0.41. The CSF made from fabric contained from 0.07% to 2% of N, and the model had R^2^ = 0.51. The CSF produced from kitchen waste contained 2.5% to 7% of N, and the model had R^2^ = 0.46. For CSF made from the paper, the N was between 0.02% and 0.93%, and the model had R^2^ = 0.69. The CSF made from plastic contains 0.06% to 7.2% of N, and the model had R^2^ = 0.13. The N of CSF made of rubber was between 0.12% and 1.72%, and the model R^2^ = 0.38. The CSF made from PAP/AL/PE composite packaging pack contained from 0.01% to 5.1% of N, and the model R^2^ = 0.21. The N in CSF from wood ranged from 0.96% to 3.8%, and model R^2^ = 0.46.

In general, all N models had low R^2^; therefore, the proposed models are not suitable for accurate prediction of N content in CSFs, but instead, they can be used to estimate the concentration range for N after pyrolysis.

The correlation between the experiment and the model data is shown in Figure 10. The results showed that the proposed model explained 0% and 7% of data variability for RDF blend 1 and RDF blend 2, respectively. The confidence interval was 95%. The results are characterized by high scatter, which makes the prediction rather not reliable. Based on the proposed model, only the general range of N is possible to check (for tested RDF N content in a narrow range of ~0.4–0.6%)

#### 3.1.11. Sulfur Content of CSF

For all CSF, the best model for S determination was model IV. These models IV had the highest R^2^ and the lowest AIC value (Table A3). The equations of influence of low-temperature pyrolysis T and t on the S of CSF are summarized in Table 11.

The S in CSF from the carton ranged from 0.01% to 0.21%, and the model had R^2^ = 0.41. The CSF made from cotton contained 0.02% to 0.27% of S, and the model had R^2^ = 0.51. The CSF produced from kitchen waste contained from 0.17% to 0.27% of S, and the model had R^2^ = 0.46. The S of CSF made from the paper was between 0.06% and 0.2%, and the model had R^2^ = 0.69. The CSF made from plastic ranged from 0.01% to 0.18% of S, and the model had low R^2^ = 0.13. S content in CSF made of rubber was from 1.66% to 2.30%, and the model had R^2^ = 0.38. CSF made from PAP/AL/PE composite packaging pack contained from 0.04% to 0.52% of S, and the model had R^2^ = 0.21. The S of CSF produced from wood ranged from 0.03% to 0.40%, and the model had R^2^ = 0.46. These results show that the pyrolysis of MSW components does not cause a significant change in its S content.

The correlation between the experiment and the model for S content is shown in Figure 11. The results showed that the proposed model explained 56% and 61% of data variability for RDF blend 1 and RDF blend 2. The confidence interval was 95%. The largest concentration of results is in 0.1% for experimental data and 0.2% for the models, which means the model overestimated experimental results by approximately a factor of 2.

## 4. Conclusions

The results show that most of the proposed general models had an R^2^ > 0.6 with experimental data. The best prediction was found for MY (R^2^ = 0.9). The presented model and the Appendix A spreadsheet with implemented equations are a simple tool to rapidly predict CSF’s expected properties made from a specific mixture of common feedstock materials found in MSW. The proposed tool can be used to find the range of pyrolysis T and residence t based on locally available MSW blends, make decisions about an optimized mix of RDF components to obtain the desired CSF quality, depending on its purpose. However, the presented results are based on experimental data from a lab-scale experiment. Therefore, they are not suitable for reuse on an industrial-scale without prior improvement/recalculations related to different plants and units’ characteristic technical parameters. Developed models predict what can be expected after carbonization at given process parameters. Nevertheless, each device has its operational parameters that impact the pyrolysis process (heating rate, cooling time, set-up temperature, residence time, feedstock size, etc.). Thus, the models need to be up-scaled and adjusted to the plan/unit-specific available equipment to obtain desired precision.

## Figures and Tables

**Figure 1 materials-14-01191-f001:**
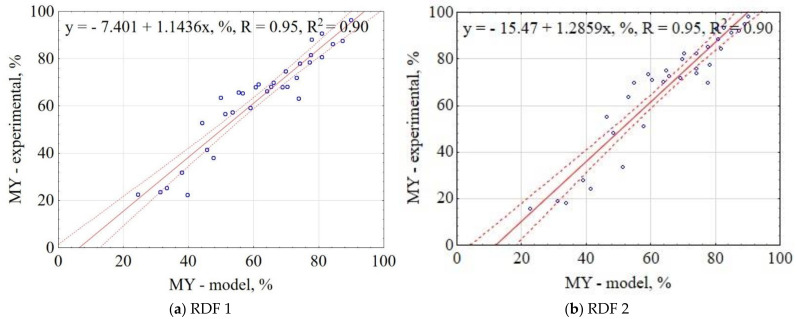
Correlation between experimental and estimated mass yield (MY) of carbonized RDF blends, (**a**) RDF 1, (**b**) RDF 2. Blue circles indicate the experimental and predicted data.

**Figure 2 materials-14-01191-f002:**
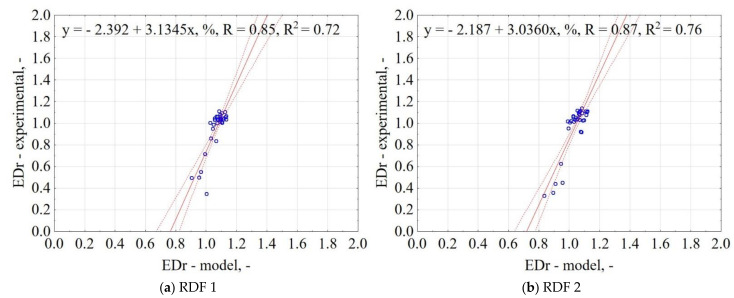
Correlation between experimental and predicted energy densification ratio (EDr) of carbonized RDF blends, (**a**) RDF 1, (**b**) RDF 2. Blue circles indicate the experimental and predicted data.

**Figure 3 materials-14-01191-f003:**
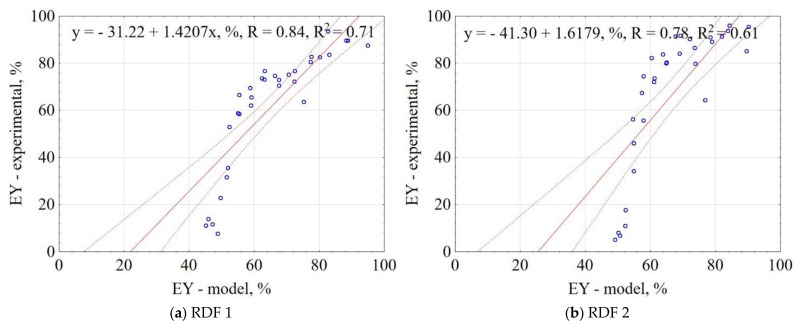
Correlation between experimental and predicted energy yield (EY) of carbonized RDF blends, (**a**) RDF blend 1, (**b**) RDF blend 2. Blue circles indicate the experimental and predicted data.

**Figure 4 materials-14-01191-f004:**
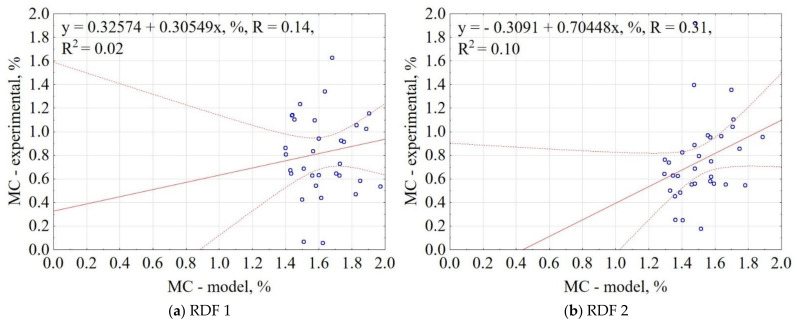
Correlation between experimental and predicted moisture content (MC) of carbonized RDF blends, (**a**) RDF blend 1, (**b**) RDF blend 2. Blue circles indicate the experimental and predicted data.

**Figure 5 materials-14-01191-f005:**
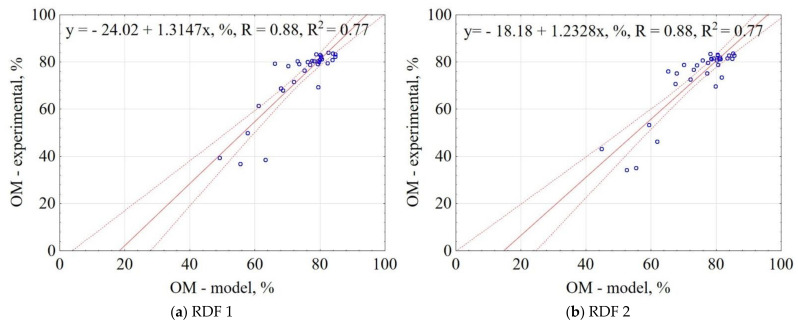
Correlation between experimental and predicted organic matter content (OM) of carbonized RDF blends, (**a**) RDF blend 1, (**b**) RDF blend 2. Blue circles indicate the experimental and predicted data.

**Figure 6 materials-14-01191-f006:**
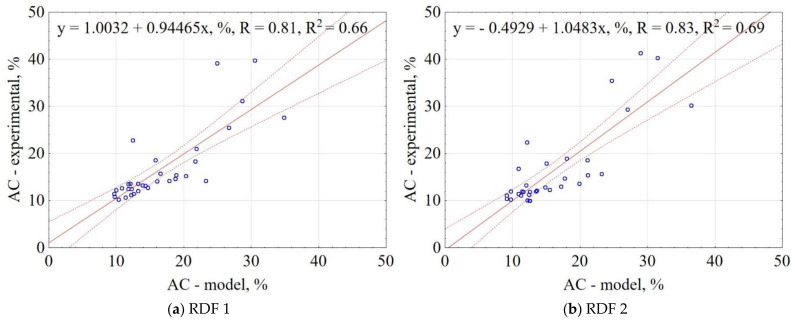
Correlation between experimental and predicted ash content (AC) of carbonized RDF blends, (**a**) RDF blend 1, (**b**) RDF blend 2. Blue circles indicate the experimental and predicted data.

**Figure 7 materials-14-01191-f007:**
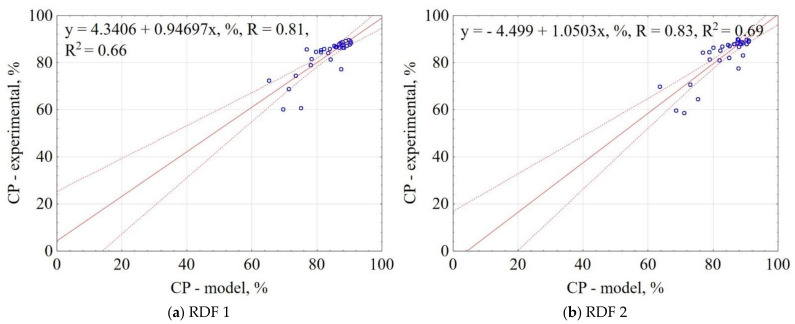
Correlation between experimental and predicted combustible parts (CP) of carbonized RDF blends, (**a**) RDF blend 1, (**b**) RDF blend 2. Blue circles indicate the experimental and predicted data.

**Figure 8 materials-14-01191-f008:**
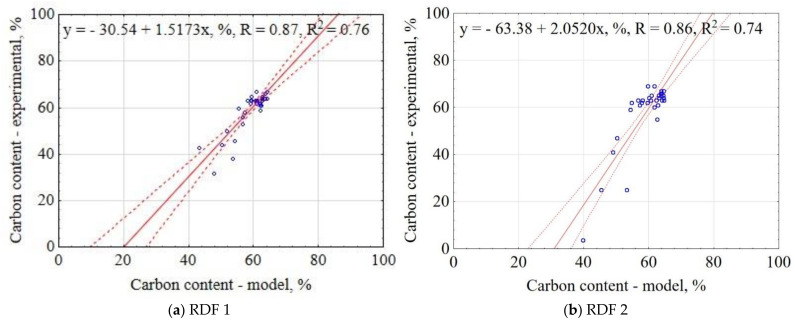
Correlation between experimental and predicted carbon content (C) of carbonized RDF blends, (**a**) RDF blend 1, (**b**) RDF blend 2. Blue circles indicate the experimental and predicted data.

**Figure 9 materials-14-01191-f009:**
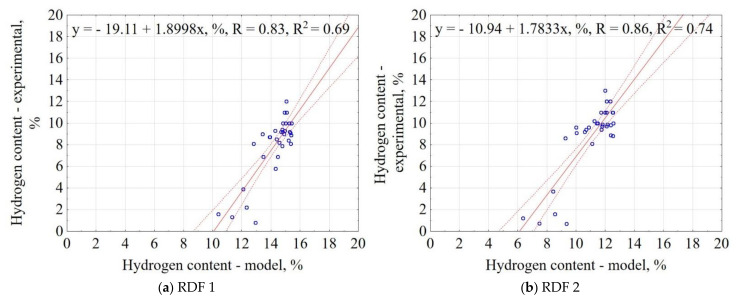
Correlation between experimental and predicted hydrogen content (H) of carbonized RDF blends, (**a**) RDF blend 1, (**b**) RDF blend 2. Blue circles indicate the experimental and predicted data.

**Figure 10 materials-14-01191-f010:**
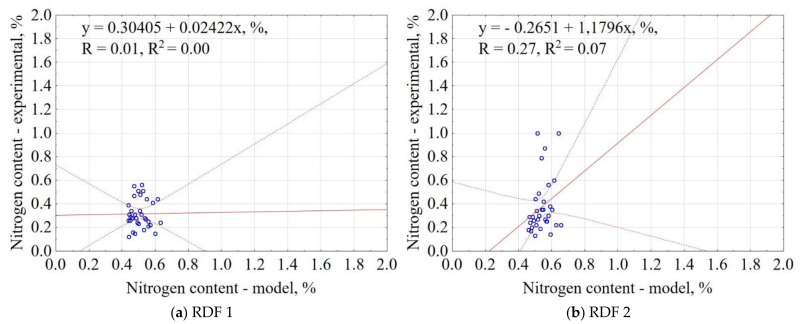
Correlation between experimental and estimated nitrogen content (N) of carbonized RDF blends, (**a**) RDF blend 1, (**b**) RDF blend 2. Blue circles indicate the experimental and predicted data.

**Figure 11 materials-14-01191-f011:**
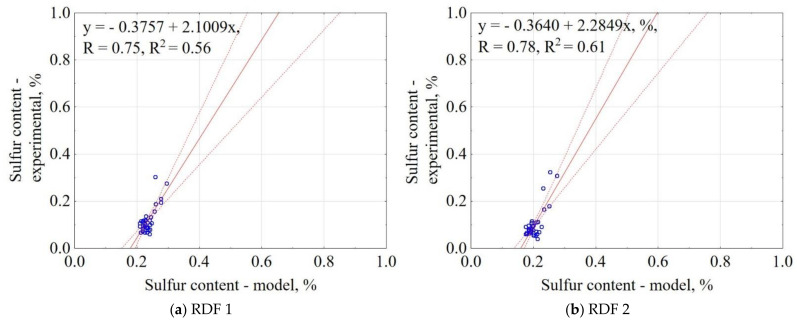
Correlation between experimental and predicted sulfur content (S) of carbonized RDF blends, (**a**) RDF blend 1, (**b**) RDF blend 2. Blue circles indicate the experimental and predicted data.

**Table 1 materials-14-01191-t001:** The chosen (best-fitted) mathematical models of the influence of the pyrolysis temperature and residence time on the mass yield of carbonized solid fuel (CSF) produced from different refuse-derived fuel (RDF) components.

Material	Equation	R^2^
Carton	MY(T,t) = 453.833 − 1.49491 × T − 3.47577 × t + 0.00137504 × T^2^ + 0.0133644 × t^2^ + 0.00499467 × T × t	0.77
Fabric	MY(T,t) = 661.088 − 2.36376 × T − 4.65892 × t + 0.00223887 × T^2^ + 0.0187874 × t^2^ + 0.00669526 × T × t	0.78
Kitchen waste	MY(T,t) = 337.338 − 1.1419 × T − 1.70984 × t + 0.00109174 × T^2^ + 0.00214157 × t^2^ + 0.00324575 × T × t	0.85
Paper	MY(T,t) = 395.322 − 1.30458 × T − 2.76693 × t + 0.00120622 × T^2^ + 0.0079086 × t^2^ + 0.00453242 × T × t	0.79
Plastic	MY(T,t) = −390.144 + 2.47977 × T + 2.62558 × t − 0.00295447 × T^2^ + 0.0075406 × t^2^ − 0.00959024 × T × t	0.72
Rubber	MY(T,t) = 216.783 − 0.27841 × T − 5.73357∙10^−^^5^ × T^2^ − 0.961876 × t + 0.00652372 × t^2^	0.85
PAP/AL/PE composite packaging pack	MY(T,t) = 268.722 − 0.611037 × T − 1.96903 × t + 0.000373715 × T^2^ + 0.0131861 × t^2^ + 0.00132244 × T × t	0.83
Wood	MY(T,t) = 341.093 − 1.08665 × T − 2.14508 × t + 0.000978011 × T^2^ + 0.00776588 × t^2^ + 0.00316288 × T × t	0.81

**Table 2 materials-14-01191-t002:** The chosen (best-fitted) mathematical models of the influence of the pyrolysis temperature and residence time on the energy densification ratio (EDr) of CSF produced from different RDF components.

Material	Equation	R^2^
Carton	EDr(T,t) = 0.521138 + 0.0024532 × T + 0.00542369 × t − 3.07672 × 10^−6^ × T^2^ − 8.29299 × 10^−5^ × t^2^ + 2.76737 × 10^−6^ × T × t	0.16
Fabric	EDr(T,t) = −0.00672325 + 0.00222108 × T + 0.0223182 × t + 1.32688 × 10^−6^ × T^2^ − 0.000176585 × t^2^ − 1.9214 × 10^−5^ × T × t	0.55
Kitchen waste	EDr(T,t) = 0.119321 + 0.00462755 × T + 0.00777896 × t − 4.13357∙10^−6^ × T^2^ + 3.05844∙10^−5^ × t^2^ − 2.39844∙10^−5^ × T × t	0.55
Paper	EDr(T,t) = 1.9716 − 0.00364838 × T − 0.00723548 × t + 3.79197 × 10^−6^ × T^2^ + 6.59619∙10^−5^ × t^2^ + 2.2538 × 10^−6^ × T × t	0.41
Plastic	EDr(T,t) = −2.65302 + 0.0183674 × T + 0.0136888 × t − 2.16719 × 10^−5^ × T^2^ + 9.6406∙10^−5^ × t^2^ − 6.10462 × 10^−5^ × T × t	0.57
Rubber	EDr(T,t) = 0.862326 + 0.00302199 × T − 0.00264502 × t − 5.80671∙10^−6^ × T^2^ + 0.000101288 × t^2^ − 2.56826 × 10^−5^ × T × t	0.88
PAP/AL/PE composite packaging pack	EDr(T,t) = −5.71804 + 0.0327167 × T + 0.0298075 × t − 3.66779 × 10^−5^ × T^2^ + 2.0243 × 10^−5^ × t^2^ − 8.0066 × 10^−5^ × T × t	0.73
Wood	EDr(T,t) = 0.0728126 + 0.0040324 × T + 0.00837479 × t − 2.97147 × 10^−6^ × T^2^ − 2.61616 × 10^−5^ × t^2^ − 1.17289 × 10^−5^ × T × t	0.82

**Table 3 materials-14-01191-t003:** The chosen (best-fitted) mathematical models of the influence of the pyrolysis temperature and residence time on the EY of CSF produced from different RDF components.

Material	Equation	R^2^
Carton	EY(T,t) = 463.255 − 1.50599 × T − 3.38498 × t + 0.00135398 × T^2^ + 0.00995947 × t^2^ + 0.00536615 × T × t	0.76
Fabric	EY(T,t) = 644.004 − 2.29865 × T − 4.12665 × t + 0.00222253 × T^2^ + 0.0151381 × t^2^ + 0.00597733 × T × t	0.81
Kitchen waste	EY(T,t) = 350.058 − 1.14163 × T − 1.55505 × t + 0.0011033 × T^2^ + 0.00316085 × t^2^ + 0.00261416 × T × t	0.81
Paper	EY(T,t) = 473.744 − 1.61526 × T − 3.26723 × t + 0.0015321 × T^2^ + 0.0111973 × t^2^ + 0.00497287 × T × t	0.81
Plastic	EY(T,t) = −9.37476 + 0.90073 × T − 0.00130925 × T^2^ − 2.83805 × t + 0.0328112 × t^2^	0.24
Rubber	EY(T,t) = 264.858 − 0.388561 × T − 5.14248∙10^−^^5^ × T^2^ − 1.56949 × t + 0.0114869∙t^2^	0.85
PAP/AL/PE composite packaging pack	EY(T,t) = −53.6102 + 1.01489 × T − 0.678159 × t − 0.00150398 × T^2^ + 0.0150335 × t^2^ − 0.00239616 × T × t	0.78
Wood	EY(T,t) = 311.037 − 0.913675 × T − 1.84836 × t + 0.00080429 × T^2^ + 0.00715877 × t^2^ + 0.00261271 × T × t	0.79

**Table 4 materials-14-01191-t004:** The chosen (best-fitted) mathematical models of the influence of the pyrolysis T and t on the moisture content (MC) of CSF produced from different RDF components.

Material	Equation	R^2^
Carton	MC(T,t) = 4.45356 − 0.0162423 × T + 0.0509884 × t + 2.81284∙10^−5^ × T^2^ + 0.000382481 × t^2^ − 0.000179923 × T × t	0.11
Fabric	MC(T,t) = −9.21332 + 0.0470837 × T − 5.06548∙10^−5^ × T^2^ + 0.0599597 × t − 0.000364179 × t^2^	0.13
Kitchen waste	MC(T,t) = −3.55952 + 0.0185074 × T + 0.116747 × t − 6.46219 × 10^−6^ × T^2^ − 0.000707797 × t^2^ − 0.000144938 × T × t	0.10
Paper	MC(T,t) = 21.8367 − 0.0859335 × T − 0.0675578 × t + 9.25108∙10^−5^ × T^2^ − 0.000173046 × t^2^ + 0.000177062 × T × t	0.21
Plastic	MC(T,t) = 8.3711 − 0.0361188 × T − 0.0326103 × t + 4.02833 × 10^−5^ × T^2^ + 0.00016858 × t^2^ + 5.13878 × 10^−5^ × T × t	0.35
Rubber	MC(T,t) = 2.14313 − 0.00478142 × T − 0.0373156 × t + 3.77281 × 10^−6^ × T^2^ + 0.000197849 × t^2^ + 6.01358 × 10^−5^ × T × t	0.05
PAP/AL/PE composite packaging pack	MC(T,t) = 29.7302 − 0.135813 × T + 0.000164348 × T^2^ − 0.053865 × t + 0.000720295 × t^2^	0.49
Wood	MC(T,t) = 0.992658 + 0.00451654 × T + 0.00649111∙t	0.05

**Table 5 materials-14-01191-t005:** The chosen mathematical models of the influence of the pyrolysis temperature and residence time on the organic matter content (OM) content of CSF produced from different RDF components are given.

Material	Equation	R^2^
Carton	OM(T,t) = 211.614 − 0.475308 × T − 1.51295 × t + 0.000401745 × T^2^ + 0.0108101 × t^2^ + 0.00120767 × T × t	0.78
Fabric	OM(T,t) = 99.7307 + 0.00951611 × T − 2.04132 × 10^−5^ × T^2^ − 0.0610921 × t + 0.000676729 × t^2^	0.31
Kitchen waste	OM(T,t) = 114.173 − 0.0560588 × T − 0.923412 × t − 9.10507 × 10^−5^ × T^2^ + 0.00277531 × t^2^ + 0.00161421 × T × t	0.49
Paper	OM(T,t) = 228.645 − 0.608106 × T − 1.07174 × t + 0.000466622 × T^2^ − 0.000843949 × t^2^ + 0.00244208 × T × t	0.72
Plastic	OM(T,t) = −370.952 + 2.24812 × T + 2.39959 × t − 0.00265716 × T^2^ + 0.00200287 × t^2^ − 0.00736942 × T × t	0.69
Rubber	OM(T,t) = 54.0888 + 0.236418 × T + 0.269375 × t − 0.000443141 × T^2^ − 0.00150753 × t^2^ − 0.00107728 × T × t	0.84
PAP/AL/PE composite packaging pack	OM(T,t) = −9.35866 + 0.604654 × T − 0.329619 × t − 0.000867447 × T^2^ + 0.00510722 × t^2^ − 0.000765698 × T × t	0.83
Wood	OM(T,t) = 104.571 − 0.00912015 × T − 1.92896 × 10^−^^5^ × T^2^ − 0.321397 × t + 0.00329522 × t^2^	0.49

**Table 6 materials-14-01191-t006:** The chosen (best-fitting) mathematical models of the influence of the pyrolysis temperature and residence time on the AC of CSF produced from different RDF components.

Material	Equation	R^2^
Carton	AC(T,t) = −95.544 + 0.392259 × T + 1.24361 × t − 0.000333039 × T^2^ − 0.00830639 × t^2^ − 0.0010555 × T × t	0.84
Fabric	AC(T,t) = 0.127194 − 0.00312043 × T + 8.32241 × 10^−^^6^ × T^2^ + 0.012343 × t − 7.1599 × 10^−5^ × t^2^	0.35
Kitchen waste	AC(T,t) = −10.3584 + 0.0434269 × T + 0.612167 × t + 2.97628 × 10^−^^5^ × T^2^ − 0.00231362 × t^2^ − 0.000874366 × T × t	0.40
Paper	AC(T,t) = −93.6731 + 0.428111 × T + 0.630943 × t − 0.000347455 × T^2^ + 0.000742667 × t^2^ − 0.00141276 × T × t	0.63
Plastic	AC(T,t) = 285.315 − 1.33859 × T − 1.72824 × t + 0.00156606 × T^2^ + 0.00119026 × t^2^ + 0.00477206 × T × t	0.77
Rubber	AC(T,t) = 31.3507 − 0.160593 × T − 0.224709 × t + 0.000313069 × T^2^ + 0.00295488 × t^2^ + 0.000553607 × T × t	0.78
PAP/AL/PE composite packaging pack	AC(T,t) = 68.7329 − 0.416816 × T + 0.333945 × t + 0.000643374 × T^2^ − 0.00512668 × t^2^ + 0.000646058 × T × t	0.86
Wood	AC(T,t) = −0.0557718 + 0.000621897 × T + 1.61345 × 10^−5^ × T^2^ + 0.120489 × t − 0.00101243 × t^2^	0.32

**Table 7 materials-14-01191-t007:** The chosen (best-fitting) mathematical models of the influence of the pyrolysis temperature and residence time on the combustible parts (CP) of CSF produced from different RDF components.

Material	Equation	R^2^
Carton	CP(T,t) = 195.544 − 0.392259 × T^−1^.24361 × t + 0.000333039 × T^2^ + 0.00830639 × t^2^ + 0.0010555 × T × t	0.84
Fabric	CP(T,t) = 99.8728 + 0.00312028 × T − 8.32223 × 10^−6^ × T^2^ − 0.012343 × t + 7.1599∙10^−5^ × t^2^	0.35
Kitchen waste	CP(T,t) = 110.358 − 0.0434266 × T − 0.612168 × t − 2.97632 × 10^−5^ × T^2^ + 0.00231363 × t^2^ + 0.000874366 × T × t	0.40
Paper	CP(T,t) = 194.858 − 0.433983 × T − 0.632625 × t + 0.000354099 × T^2^ − 0.000849486 × t^2^ + 0.0014323 × T × t	0.63
Plastic	CP(T,t) = −185.315 + 1.33859 × T + 1.72824 × t − 0.00156606 × T^2^ − 0.00119026 × t^2^ − 0.00477206 × T × t	0.77
Rubber	CP(T,t) = 68.6493 + 0.160593 × T + 0.224709 × t − 0.000313069 × T^2^ − 0.00295489 × t^2^ − 0.000553607 × T × t	0.78
PAP/AL/PE composite packaging pack	CP(T,t) = 31.2671 + 0.416816 × T − 0.333945 × t − 0.000643374 × T^2^ + 0.00512668 × t^2^ − 0.000646058 × T × t	0.86
Wood	CP(T,t) = 100.096 − 0.000577691 × T^−1^ × 62655∙10^−5^ × T^2^ − 0.122004 × t + 0.00102758 × t^2^	0.32

**Table 8 materials-14-01191-t008:** The chosen (best-fitting) mathematical models of the influence of the pyrolysis temperature and residence time on the C of CSF produced from different RDF components.

Material	Equation	R^2^
Carton	C(T,t) = 14.9394 + 0.110664 × T + 0.27 × t − 9.22688 × 10^−5^ × T^2^ + 0.001 × t^2^ − 0.000734091 × T × t	0.46
Fabric	C(T,t) = −139 + 0.776403 × T + 1.32727 × t − 0.000759518 × T^2^ − 0.00727273 × t^2^ − 0.00154545 × T × t	0.62
Kitchen waste	C(T,t) = 111.158 − 0.329169 × T + 0.342273 × t + 0.00041453 × T^2^ − 0.00179546 × t^2^ − 0.000393182 × T × t	0.26
Paper	C(T,t) = 43.697 − 0.0542969 × T + 0.429545 × t + 0.00010878 × T^2^ − 0.000568175 × t^2^ − 0.000886363 × T × t	0.24
Plastic	C(T,t) = −207.545 + 1.34185 × T + 2.12727 × t − 0.0015472 × T^2^ + 9.26989 × 10^−10^ × t^2^ − 0.00609091 × T × t	0.72
Rubber	C(T,t) = 118.755 − 0.131543 × T + 0.118908 × t + 6.15577 × 10^−5^ × T^2^ − 0.00221363 × t^2^ − 0.000437044 × T × t	0.69
PAP/AL/PE composite packaging pack	C(T,t) = −144.424 + 0.858967 × T + 1.63864 × t − 0.000925602 × T^2^ − 0.00511364 × t^2^ − 0.00310227 × T × t	0.63
Wood	C(T,t) = 43.6364 − 0.0147981 × T + 0.523864 × t + 0.000107323 × T^2^ − 0.00073864 × t^2^ − 0.000903409 × T × t	0.72

**Table 9 materials-14-01191-t009:** The chosen (best-fitting) mathematical models of the influence of the pyrolysis temperature and residence time on the H content of CSF produced from different RDF components.

Material	Equation	R^2^
Carton	H(T,t) = 25.7364 − 0.0817727 × T − 0.156818 × t + 7.95455 × 10^−5^ × T^2^ + 0.00120455 × t^2^ + 8.63636 × 10^−5^ × T × t	0.92
Fabric	H(T,t) = 34.3758 − 0.114567 × T − 0.164091 × t + 0.000107906 × T^2^ + 6.81809 × 10^−5^ × t^2^ + 0.000285227 × T × t	0.79
Kitchen waste	H(T,t) = 15.6939 − 0.041657 × T + 3.73932 × 10^−5^ × T^2^ − 0.0625 × t + 0.000534091 × t^2^	0.83
Paper	H(T,t) = 23.429 − 0.0742717 × T − 0.145876 × t + 6.72703 × 10^−5^ × T^2^ + 0.000605284 × t^2^ + 0.000175301 × T × t	0.88
Plastic	H(T,t) = −44.8006 + 0.276956 × T + 0.432934 × t − 0.000314235 × T^2^ + 0.00050216 × t^2^ − 0.00135652 × T × t	0.66
Rubber	H(T,t) = 26.1644 − 0.0922177 × T − 0.177385 × t + 9.00828 × 10^−5^ × T^2^ + 0.00123522 × t^2^ + 0.000127838 × T × t	0.91
PAP/AL/PE composite packaging pack	H(T,t) = −20.7405 + 0.15232 × T + 0.0595576 × t − 0.000193613 × T^2^ + 0.000570871 × t^2^ − 0.000377273 × T × t	0.72
Wood	H(T,t) = 4.36363 − 0.00147975 × T + 0.0523864 × t + 1.07323 × 10^−5^ × T^2^ − 7.38636 × 10^−5^ × t^2^ − 9.03409 × 10^−5^ × T × t	0.72

**Table 10 materials-14-01191-t010:** The chosen (best-fitting) mathematical models of the influence of the pyrolysis temperature and residence time on the N content of CSF produced from different RDF components.

Material	Equation	R^2^
Carton	N(T,t) = 0.353788 − 0.00163594 × T + 0.0102094 × t + 2.79794 × 10^−^^6^ × T2 − 4.875 × 10^−^^5^ × t^2^ − 1.24001 × 10^−^^5^ × T × t	0.41
Fabric	N(T,t) = −0.538518 − 0.00249781 × T + 0.0363215 × t + 1.09306 × 10^−5^ × T^2^ − 0.000272625 × t^2^ − 3.04533 × 10^−5^ × T × t	0.51
Kitchen waste	N(T,t) = 7.29638 − 0.0187967 × T + 0.0120734 × t + 2.08782 × 10^−5^ × T^2^ + 8.75 × 10^−5^ × t^2^ − 4.88262 × 10^−5^ × T × t	0.46
Paper	N(T,t) = 0.56423 − 0.00232121 × T + 0.000358041 × t + 2.63612 × 10^−6^ × T^2^ + 8.12501 × 10^−6^ × t^2^ − 1.61463 × 10^−6^ × T × t	0.69
Plastic	N(T,t) = −1.5433 + 0.00661536 × T + 0.035132 × t − 8.45851 × 10^−6^ × T^2^ − 0.000255625 × t^2^ − 2.67845 × 10^−5^ × T × t	0.13
Rubber	N(T,t) = 0.183955 − 0.000655605 × T + 0.0100913 × t + 1.64191 × 10^−6^ × T^2^ − 3.25 × 10^−5^ × t^2^ − 1.45105 × 10^−5^ × T × t	0.38
PAP/AL/PE composite packaging pack	N(T,t) = 0.277347 − 0.00120315 × T + 0.00538399 × t + 1.20488 × 10^−6^ × T^2^ − 5.275 × 10^−5^ × t^2^ − 2.4975 × 10^−6^ × T × t	0.21
Wood	N(T,t) = −0.743849 + 0.00728439 × T + 0.0366098 × t − 7.02211 × 10^−6^ × T^2^ + − 6.75002 × 10^−5^ × t^2^ − 6.24626 × 10^−5^ × T × t	0.46

**Table 11 materials-14-01191-t011:** The chosen (best-fitting) mathematical models of the influence of the pyrolysis temperature and residence time on the S content of CSF produced from different RDF components.

Material	Equation	R^2^
Carton	S(T,t) = −0.177334 + 0.000937873 × T + 0.00420794 × t − 4.95826 × 10^−7^ × T^2^ − 8.52128 × 10^−7^ × t^2^ − 8.69886 × 10^−6^ × T × t	0.26
Fabric	S(T,t) = 1.24721 − 0.00598883 × T + 0.00300454 × t + 7.38831∙10^−6^ × T^2^ − 1.59088 × 10^−6^ × t^2^ − 7.19318 × 10^−6^ × T × t	0.39
Kitchen waste	S(T,t) = 0.76394 − 0.00268124 × T − 0.000843186 × t + 3.00117 × 10^−6^ × T^2^ − 4.43179 × 10^−6^ × t^2^ + 3.75 × 10^−6^ × T × t	0.35
Paper	S(T,t) = 0.852576 − 0.00438192 × T + 0.00400454 × t + 5.96737 × 10^−6^ × T^2^ + 9.09103 × 10^−7^ × t^2^ − 8.71591 × 10^−6^ × T × t	0.49
Plastic	S(T,t) = 1.04115 − 0.00503069 × T − 0.00623409 × t + 5.96639 × 10^−6^ × T^2^ + 5.68193 × 10^−7^ × t^2^ + 1.75341 × 10^−5^ × T × t	0.78
Rubber	S(T,t) = 1.3005 + 0.00161075 × T + 0.00576999 × t − 1.16336 × 10^−6^ × T^2^ − 6.05901 × 10^−6^ × t^2^ − 1.13023 × 10^−5^ × T × t	0.28
PAP/AL/PE composite packaging pack	S(T,t) = 0.672402 − 0.00299589 × T − 0.00228128 × t + 3.56719 × 10^−6^ × T^2^ − 1.87036 × 10^−6^ × t^2^ + 6.33409 × 10^−6^ × T × t	0.61
Wood	S(T,t) = 0.278057 − 0.00127582 × T + 0.00100456 × t + 1.59488 × 10^−6^ × T^2^ − 6.11576 × 10^−6^ × t^2^ − 1.04858 × 10^−6^ × T × t	0.33

## Data Availability

The data used in this study come from work Świechowski, K.; Syguła, E.; Koziel, J.A.; Stępień, P.; Kugler, S.; Manczarski, P.; Białowiec, A. Low-Temperature Pyrolysis of Municipal Solid Waste Components and Refuse-Derived Fuel—Process Efficiency and Fuel Properties of Carbonized Solid Fuel. Data 2020, 5, 48. https://doi.org/10.3390/data5020048 (accessed on 25 February 2021) and are available in https://www.mdpi.com/2306-5729/5/2/48/s1 (accessed on 25 February 2021) and data generated in this study are available in the Appendix A named “Supplementary Materials-Models”.

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
