# Peer review of "Lab-Scale Study of Temperature and Duration Effects on Carbonized Solid Fuels Properties Produced from Municipal Solid Waste Components"

_materials, 2021, doi:10.3390/ma14051191_

Round 1
Reviewer 1 Report
This paper investigates the model of CSF from different temperature and time. It is not a typical material article, since it focuses on statistics. The only concern is whether the content in this article is within the scope of the journal. I will leave this question to the editor. Another question from me is: It is not clear that why CO2 is chosen as the pyrolysis gas. Pure CO2 is not easily obtained compared to N2.
What is the main question addressed by the research? The paper addressed the modelling of carbon product from MSW pyrolysis at different conditions.
Is it relevant and interesting? I think it is not fully in the scope of Materials. It is more about chemical engineering or processes.
How original is the topic? It has a clear novelty.
What does it add to the subject area compared with other published material? It will help to predict the carbon product simulation from MSW pyrolysis.
Is the paper well written? Yes
Is the text clear and easy to read? Yes
Are the conclusions consistent with the evidence and arguments presented? Yes
Do they address the main question posed? Yes.
Author Response
The responses to the reviewer's comments are in the attached file.

Reviewer 2 Report
The paper entitled “Pyrolysis in Carbon Dioxide: Effects of Temperature and Duration on Carbonized Solid Fuels Properties Produced from Municipal Solid Waste Components” Kacper Åšwiechowski, PaweÅ‚ StÄ™pieÅ„, Ewa SyguÅ‚a, Jacek A. Koziel and Andrzej BiaÅ‚owiec, appears to be incomplete in the analyzes, therefore it needs major revisions to be published in Materials.
The most important points, which the authors should necessarily review in the paper, are clarified below.
1) The title does not exactly reflect the paper content. The title should stress that the research is conducted on a laboratory scale. Furthermore, CO2 should also not be considered in the title because it was used only as an inertizer and its possible effect on pyrolysis reactions is not studied.
2) In the introduction, the authors should precisely frame the paper with respect to the different pyrolysis models reported in the literature. In particular, the actors should indicate what are the innovative aspects of their model.
3) Authors must specify exactly which statistical methodologies were used and, above all, justify their choice.
4) The authors should show the reader whether the data and experimental conditions used meet the applicability limits of the chosen statistical analysis methodologies.
5) In sections 2.2.2 "Regression Modeling" and 2.2.3 "General Model", the authors should show the number of experiments conducted and the corresponding replicates as well as the verification tests necessary for the correct application of the statistical methodologies, etc.
6) Figures 1, 3, 5, 7, 9,11,13, 15, 17, 19, 21, should be converted into tables containing only the polynomial equations and the related statistical parameters that characterize them.
7) Authors should clarify the limitations of the methods used in the laboratory, but should show the reader how their results can be scaled to translate them into the practice of an industrial pyrolyzer.
Author Response

(The authors gave the same response as above.)

Reviewer 3 Report
-The present work has redundant data. It's so hard to follow the context from the point of the reader's view. 23 figures are too much. They should select the main figures and rewrite the context.
Author Response

(The authors gave the same response as above.)

Round 2
Reviewer 2 Report
This version of the manuscript is much improved, however for its publication it is necessary that the authors delete the figures in the appendix. Indeed, they are a repetition of what is already reported in the tables and in the text.
Reviewer 3 Report
Better than the previous version.
Still, the section of the conclusion is too large. What is the main message?
